# Birth Weight and Body Composition as Determined by Isotopic Dilution with Deuterium Oxide in 6- to 8-Year-Old South African Children [note 1]

**DOI:** 10.3390/children9101597

**Published:** 2022-10-21

**Authors:** Makama Andries Monyeki, Caroline Molete Sedumedi, John J. Reilly, Xanne Janssen, Herculina Salome Kruger, Ruan Kruger, Cornelia U. Loechl

**Affiliations:** 1Physical Activity, Sport and Recreation Research Focus Area (PhASRec), Faculty of Health Sciences, North-West University, Potchefstroom 2520, South Africa; 2Physical Activity for Health Group, School of Psychological Sciences and Health, University of Strathclyde, Glasgow G1 1QE, UK; 3Centre of Excellence for Nutrition, North-West University, Potchefstroom 2520, South Africa; 4Medical Research Council (MRC) Research Unit for Hypertension and Cardiovascular Disease, North-West University, Potchefstroom 2520, South Africa; 5Hypertension in Africa Research Team (HART), North-West University, Potchefstroom 2520, South Africa; 6International Atomic Energy Agency (IAEA), Division of Human Health, Vienna International Centre, 1400 Vienna, Austria

**Keywords:** birth weight, body composition, obesity, stable isotope, South African children

## Abstract

Low and high birth weight (BW) are associated with obesity later in life; however, this association has not been extensively studied in African countries. This study determines the association between BW and body composition derived from deuterium oxide (D_2_O) dilution in 6- to 8-year-old South African children (*n* = 91; 40 boys, 51 girls). BW was recorded retrospectively from the children’s Road-to-Health cards. Weight and height were measured using standard procedures, and D_2_O dilution was used to determine total body water and, subsequently, to determine body fat. Fatness was classified using the McCarthy centiles, set at 2nd, 85th, and 95th (underfat, overfat and obese). BW correlated with body composition measures, such as body weight (r = 0.23, *p* = 0.03), height (r = 0.33, *p* < 0.001), and fat free mass (FFM; r = 0.27, *p* = 0.01). When multiple regression analysis was employed, BW significantly and positively associated with FFM (β = 0.24, *p* = 0.013; 95% CI: 0.032; 0.441) and fat mass (β = 0.21, *p* = 0.02, 95%CI: 0.001; 0.412) in girls and boys combined. A total of 13% of the children had a low BW, with 21% being overweight and 17% obese. More girls than boys were overweight and obese. Intervention strategies that promote healthy uterine growth for optimal BW are needed in order to curb the global obesity pandemic.

## 1. Introduction

Childhood obesity is increasing at an alarming rate globally, with approximately 50 million girls and 75 million boys affected [1]. Childhood obesity is associated with an increased risk of obesity in adulthood and early life onset of non-communicable diseases [2,3]. A child’s risk for obesity begins with genetic factors inherited from his or her parents [4]. Disturbances that occur in the uterus or in infancy may cause longstanding metabolic and physiological adaptations that result in obesity later in life. Such factors include maternal over-nutrition, diabetes mellitus, and excessive gestational weight gain [5,6].

The risk of obesity is further increased by environmental factors such as low physical activity, poor diet, socio-economic status (SES), and breastfeeding practices [4,7,8]. Although paucity exists in the data, maternal malnutrition also has an increased risk on the development of obesity in the child later in life [4].

Birth weight (BW) has been used as an indicator of intrauterine growth [9]. High birth weight (HBW) is associated with obesity in childhood [4,10]. Foetal undernutrition results in low birth weight (LBW), increasing the risk of obesity later in life when associated with rapid postnatal catch-up growth [7], and a higher risk of chronic diseases such as cardiovascular disease, diabetes mellitus, and poor liver function [11,12,13,14]. Most studies that have assessed the association between BW and overweight or obesity in childhood used the body mass index (BMI) as a proxy for body fatness [6,10,15]. Such studies have reported that HBW is associated with high BMI; however, high BMI could either be a result of high fat mass (FM) or of high fat free mass (FFM) [6,16]. BMI with appropriate cut-points performed reasonably well in detecting those with high body fatness in African children; however, it shows only moderate sensitivity in differentiating between normal and high body fatness [17,18].

The need to report body composition using FM and FFM, especially in children, is of paramount importance. Yet, inconsistent findings have been reported on the impact of BW on both FM and FFM [4,19]. Bernhardsen and colleagues [20] reported no relation between BW and body composition (FM and FFM) in Norwegian children aged 9–12 years, whereas Pruszkowska-Przybylska and colleagues [21] reported a relationship between HBW and high BMI, and low FFM percentage (FFM%) in 6- to 13-year-old Polish children. Birth weight was found to be positively associated with FM and FFM in 5- to 8-year-old Brazilian children [22].

It is, therefore, not known whether similar associations between birth weight and body composition are present in South African children, who are faced with the paradoxes of the nutritional transition linked to both underweight and obesity. It was against this background that we aimed to determine the association between BW and body composition as determined by D_2_O in 6- to 8-year-old South African children. We hypothesized that BW would be positively associated with the body composition components of FM and FFM in childhood. This information will be useful in designing strategies for improving maternal health status in order to prevent either obesity or underweight in childhood.

## 2. Materials and Methods

### 2.1. Study Design and Participants

In this study, a cross-sectional descriptive study design was used in a sub-sample of 91 children (40 boys and 51 girls) from a larger study on body composition, using a stable isotope technique (BC–IT study) [23,24]. The 91 participants were included because they had complete retrospective BW (primary exposure) data recorded from their Road-to-Health cards (RtHC) (Figure 1). The RtHC is designed to keep track of babies’ healthcare needs, and is free for all babies born in South Africa at public or private healthcare facilities. The RtHC is a useful databank and patient-held child medical record, as it summarises a child’s health status in the first five years of life [25], and it depends on the knowledge, dedication, and co-operation of mothers or caregivers and healthcare personnel [26]. The details about the sample size in this subsample of 91 participants have been published elsewhere [24]. This sample size was, however, largely dependent on the availability of a completed RtHC birth record for each child and should not be viewed as a representation of children in the age groups. More details about the BC–IT study have been published elsewhere [23,24].

Permission for the study was received from the school principals and district office of the Department of Basic Education in the Dr Kenneth Kaunda District Municipality of North West Province. Subsequently, the Human Research Ethics Committee in the Faculty of Health Sciences (HREC) of North-West University approved the study protocol (ethics no: NWU-00025-17-A1). Participant recruitment was conducted through advertisement in primary schools in the district, with permission from the Department of Education. A sample that best represented both high and low SES was selected from the schools within Potchefstroom in the North West province of South Africa. The class lists for Grade R to Grade 3 were requested from the five participating schools, and every third child on each class list was nominated to participate in the study. Subsequently, only children with parental informed consent and verbal assent (6-year-olds)/written consent (7–8-year-olds) were included in the study. Furthermore, in this sub-sample, only children who provided the completed RtHC were included. Data collection started in September 2017 and lasted until May 2019.

### 2.2. Measuring Instruments

#### 2.2.1. Socio-Demographic Questionnaire

Information on families’ socioeconomic status (SES) (i.e., education of the parents, employment, type of dwelling, household amenities and marital status), were collected by trained fieldworkers, as previously described [24].

#### 2.2.2. Birth Weight

For each participating child, BW was retrieved from the RtHC and recorded in kilograms (kg). For analysis, BW was then categorised into low birth weight (LBW; <2.5 kg), normal birth weight (NBW; ≥2.5 kg to <4.5 kg) and high birth weight (HBW; ≥4.5 kg), in line with the World Health Organization (WHO) definitions [27,28,29].

#### 2.2.3. Anthropometric Measurements

Body height (cm), weight (kg), and waist circumference (cm) were determined by following the International Society for the Advancement of Kinanthropometry (ISAK) procedures [30]. Measurements were conducted by Level I anthropometrists, and boys and girls were measured in separate rooms to ensure privacy [24]. The BC–IT study measurements procedures were described elsewhere [23,24]. Briefly, body height was measured to the nearest 0.1 cm by a Seca 213 stadiometer (Birmingham, UK) with a child standing barefoot and upright with their head in the Frankfort plane position. Additionally, body weight to the nearest 0.1 kg was measured using a Seca 813 digital scale (Birmingham, UK). A Lufkin metal tape (Cooper Industries, Texas, USA) was used to measure waist circumference to the nearest 0.1 cm at the midpoint between the lower rib margin and the iliac crest.

BMI z-scores were calculated using WHO reference data, and subsequently, as previously described in the BC-IT study [23], participants were categorised as follows: underweight: <−2 standard deviations (SD) from the median; normal weight: −2 SD to +1 SD; overweight: more than +1 to +2 SD; and obesity: more than +2 SD [31].

#### 2.2.4. Body Composition by Isotopic Dilution with Deuterium (D_2_O) Oxide

The determination of FM% by D_2_O was guided by the protocol provided by the International Atomic Energy Agency (IAEA) [32]. The implementation of the current study follows a multi-centre study, which involves twelve African countries [33]. After an overnight fast, children provided a saliva sample, referred to as the pre-dose saliva sample. Each child was given a dose of accurately prepared deuterium oxide-labelled water according to their body weight and age. The doses were administered under close supervision with drinking straws to avoid potential spillage. In order to ensure the complete intake of the dose, dose bottles were rinsed with water (50 mL per rinse) twice and the rinsing water was consumed by the child. The time for dose administration was recorded and, at two and three hours post dosage, saliva samples were collected. In the period of dose administration and sample collection, children were asked to stay in the study location and no physical activity was permitted, in order to minimize water loss in breath and evaporation from the skin. We, therefore, provided children with non-vigorous activities, such as colouring books and storytelling, to prevent boredom during the waiting period until the study protocol was completed. No eating or drinking was permitted apart from the light snack that was provided after completion of the sample collection. Children, under supervision, were permitted to go to the bathroom at any time. Three plastic vials were used to collect three sets of saliva samples for each child (1 pre-dose, and 2 post dose) and were immediately capped to avoid evaporation. The plastic vial was encased in a sealable bag preventing possible losses from evaporation and cross-contamination during storage. The saliva samples were stored at −20 °C in the lab until analysis was performed. Fourier transform infrared (FTIR) spectroscopy (FTIR 4500t spectrophotometer, Agilent) was used to analyse the saliva samples. MicroLab PC software was used to obtain valuable information about the identity and amount of chemical substances present in a material. This D_2_O method has precision and accuracy rates of 1–2% for measuring TBW. The hydration factor is assumed to be 73%, and the FFM is estimated from TBW using this assumption. Since the percentage of water in the FFM was found to be between 70% and 76% in most species, it would be ideal to use population-specific hydration factors of FFM. The estimates of %BF from the TBW method showed small differences (<1% BF) as compared to the 4-compartment model [32,33,34]. In order to determine FFM, age- and sex-specific Lohman hydration factors for children were used, and 4 FM and FM% were calculated [32,33,34]. McCarthy et al. [35] centiles set at the 2nd, 85th, and 95th, respectively, were used to classify the children as underfat/underweight, overfat/overweight, and obese. Normal fatness in the study was set at above the 9th to below the 85th centile [33,35].

### 2.3. Statistical Analysis

Data were analysed using the Statistical Package for the Social Sciences (SPSS) software version 26 (IBM Corp., Armonk, NY, USA). Normality was assessed by visual inspection of QQ plots and the statistical one-sample Kolmogorov–Smirnov normal distribution examination. Logarithmic transformations were performed for data that were not normally distributed (BMI z-score, FM). In order to describe participants’ characteristics, descriptive statistics in terms of means and standard deviations were calculated for continuous data. Independent Welch’s unequal variances t-tests were performed to determine the differences between the birth weight categories. For all categorical variables, percentages were calculated, and Pearson’s Chi-square (χ2) was calculated to determine the differences between categories. Multiple regression analysis (‘crude,’ and adjusted for age, sex, and SES factor for family income) and partial correlation were determined in order to show the relationship between BW (independent variable) and childhood body composition (dependent variables). In performing regression models and summarizing data in a correlation matrix showing correlation coefficients between variables, analyses were adjusted for age, sex, and selected socio-economic factors. The probability level of significance was set at *p* ≤ 0.05.

## 3. Results

A total of 91 children (40 boys and 51 girls) with birth weight data were included. Table 1 presents the descriptive statistics of the participants in terms of means, minimum, maximum, and SDs.

Thirteen percent (13%) of the total participants were born with LBW. The prevalence of overweight and obesity was 21% and 17%, respectively, when using the D_2_O based on body fatness (Table 2). In contrast, the prevalence of overweight and obesity was 13% and 5%, respectively, according to BMI z-score. Girls showed a higher prevalence of overweight and obesity compared with boys when the D_2_O was used, but there were no significant differences based on BMI z-scores.

Table 3 presents descriptive statistics according to birth weight groups. Children with NBW were significantly taller than the LBW group. Although not significant, children with LBW showed a trend of being smaller than children with NBW, with a strong trend of lower FM in the LBW group compared with the NBW group (*p* = 0.08).

Table 4 presents the associations between BW and body composition measures, as well as partial correlation, controlled for age, sex, and selected SES variables. Before adjustments were made for confounding factors, BW showed a positive correlation with weight (r = 0.23, *p* = 0.03), height (r = 0.33, *p* < 0.001), FFM (r = 0.27, *p* = 0.01), and FM (r = 0.23, *p* = 0.03). After adjustment of age and sex, only height showed a significant correlation with BW. This correlation was no longer significant after adjustment for parent education level and household income.

Table 5 shows a multiple regression analysis model, crude and adjusted. In the crude model, BW was positively associated with FFM (β = 0.24, *p* = 0.013; 95%CI: 0.032; 0.441) and FM (β = 0.21, *p* = 0.02, 95%CI: 0.001; 0.412) in girls and boys combined. When the group was separated according to sex, BW was positively associated with FFM (β = 0.32, *p* = 0.02, 95%CI: 0.050; 0.574) and showed a trend of positive association with FM (β = 0.27, *p* = 0.054) in girls. After adjustments were made for age, sex, and household income in the total group, only FM remained positively associated with BW (β = 0.23, *p* = 0.04, 95%CI: 0.031; 0.375); however, sex (*p* = 0.007) showed a positive association with FM in the same model. In the adjusted regression model for girls and boys separately, BW explains a very low percentage of variance in the body composition variables.

## 4. Discussion

The aim of the study was to determine the association between BW and body composition in 6- to 8-year-old South African children. In summary, the results from a sample of children from Potchefstroom in the North West province of South Africa showed that BW was significantly and positively associated with body composition measures of FFM and FM. LBW was reported in 13% of the children, and children with LBW were significantly smaller in terms of height compared with children with NBW. A high prevalence of overweight and obesity was reported in children, with girls more affected than boys.

The positive association between BW and body composition in childhood has been shown in several studies [11,21,36]. In the current study, a trend of a positive association between BW and FM is reported even after adjustment for possible confounders. FFM and FM were not significantly lower in the LBW group compared to the NBW group (Table 3), but there was a trend of lower FM in the LBW group (*p* = 0.08). The small number of children in the LBW group (*n* = 12) is probably the reason why statistically significant differences could not be detected. In addition, it is noteworthy to mention that sex was the strongest predictor of FM in this age group. The observed significant association between BW and FM in our study is somewhat different from other studies. To cite examples, Bernhardsen and colleagues [20] did not demonstrate any relationship between BW and body composition (FM and FFM) in 9- to 12-year-old children. No significant relationship was reported between BW and FM in European children and adolescents, although a positive relationship with FFM index was reported, but in boys only [36]. A longitudinal study in 7- to 10-year-old Brazilian children also reported no significant associations [37]. The mechanism that drives the differential development of FM and FFM in the uterus is unclear; however, it has been suggested that maternal nutrition and hormones are key contributors [11]. This highlights the impact of maternal factors on the association between birth weight and body composition. Care should be taken to ensure optimum maternal nutrition and a healthy intrauterine environment in order to prevent obesity and related health risk factors in early childhood. The observed differences in the relationships between BW and body composition variables in our study and the literature may in part be explained by the different methods used to determine body composition.

The prevalence of overweight and obesity in children is rising at an alarming rate in Sub-Saharan Africa [38]. Similarly, the present study reports a high percentage of overweight and obesity in children, especially in girls; however, we failed to show conclusively its significant association with birth weight at the age of 6 to 8 years. Birth weight may show a weaker association with body composition in the present study as a result of the small number of children with LBW and no children with HBW. The observed contradictory findings may also be explained in part by the different methods used in determining body composition and the different cut-points applied to classify body composition compared to other studies. The age of the children when outcomes are assessed may also play a role. Aside from methodological differences, there are several confounding factors that might have an effect on childhood body composition. Diet, physical activity, sleep, familial factors such as SES, genetics, and maternal factors, to name a few, have been linked to childhood obesity [39,40]. In the present study, sex and SES showed stronger associations with childhood body composition than BW. Unlike in developed countries, where high SES and childhood obesity show an inverse relationship, SES in Sub-Saharan African countries is one of the driving forces behind the development of obesity [39,41]. More attention should be given to children with high SES, particularly girls, as they appear to be more affected by childhood obesity.

Intrauterine growth indicated by BW is one of the factors that drives the global epidemic of obesity [39]. The noted prevalence of LBW (13%) is a concern for future growth and development of the children in the study, for several reasons. Physiological reasons underpinning LBW might be an indication of lower levels of androgens, thus generating positive relationships between BW and muscle mass [11,42,43,44]. LBW may increase the risk of diabetes mellitus if it results in low FFM [11,45]. The results of the current study, however, did not show any significant differences in FFM between the LBW group and the NBW group. Furthermore, there was no significant difference in FM between the LBW and NBW groups; the difference was only noted in height, with children with LBW being significantly shorter.

The strength and novelty (i.e., South African context) of this study is that it reports body composition in terms of FM and FFM determined by the D_2_O, in contrast to many previous studies, which have used the more crude outcome variable of BMI-for-age. This allowed us to examine the association between BW and body composition using objective measures in South African children. Findings from this study can be used to guide larger studies on which method to use to measure adiposity. However, there are limitations, for example, missing retrospective BW from RtHC from the larger BC–IT study affected the sample size of the sub-study, and might have influenced the result, particularly the finding of no difference in FFM and FM between the LBW and NBW groups. The use of retrospective BW has been acknowledged by some researchers [46] as a significant challenge in South Africa and other Sub-Saharan countries [47], which is due to children been born outside the health facility or loss of the RtHC. Another notable factor is that this study was based on a cross-sectional study design, and, therefore, no causal relationship can be suggested. From the available literature, the use of BW as a proxy for prenatal growth has some limitations [37], due the fact that body composition may vary even when BW is the same and within the normal range [48]. To cite an example, in a study conducted by Yajnik and co-workers [48], it was revealed that Indian babies have higher FM and lower FFM, and were therefore “small and thin” relative to UK infants. The lack of data on the gestational age, and mother’s maternal dietary practice, both of which are linked to body height, body weight, FFM, and FM, are limitations of the study. The study sample was not representative, and the results cannot be generalised to the population of the North West province, nor the entirety of South Africa. Additionally, a lack of information on parental body size, breastfeeding practices, and mothers’ smoking habits during pregnancy limited the adjustment of the observed associations in the study.

## 5. Conclusions

In line with our hypothesis, BW was significantly associated with FFM and FM at age 6 to 8 years. However, sex appeared to be the strongest predictor in the relationship between BW and FM in this study. The prevalence of overweight and obesity was higher and more pronounced in girls compared to boys. As such, sex-specific intervention strategies that will help promote healthy body composition and normal BW are needed in order to curb the obesity pandemic.

## Figures and Tables

**Figure 1 children-09-01597-f001:**
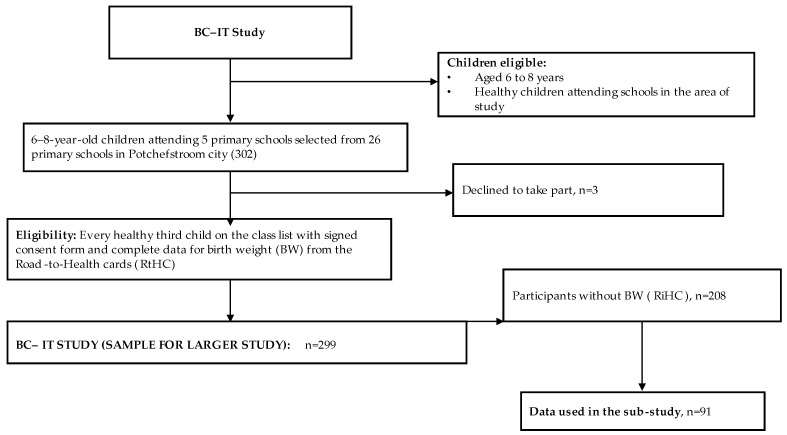
Flow diagram for the larger body composition using the isotope technique (BC-IT study) study.

**Table 1 children-09-01597-t001:** Descriptive statistics (minimum, maximum, mean, and SD) and percentages (%) for categorical variables of participants.

Variables	N	Minimum	Maximum	Mean	SD
Age (Years)	91	6.0	8.8	7.7	0.7
BW (g)	91	1000	4320	3053.9	538.2
Weight (kg)	91	15.1	48.7	25.0	5.9
Height (cm)	91	103.5	138.4	122.2	7.0
FFM (kg)	91	10.7	25.9	17.7	3.2
FM (kg)	91	2.7	22.8	7.3	3.5
FM (%)	91	12.0	46.8	28.1	7.2
TBW (*ℓ*; D_2_O)	91	8.3	19.9	13.7	2.4
BMI for age Z-score	91	−2.4	3.6	0.3	1.2

BMI = Body mass index; cm = centimetres; FFM = fat free mass; FM = fat mass; FM(%) = fat mass percentage; kg = kilograms, D_2_O = Deuterium Oxide, N = number; *ℓ* = liter.

**Table 2 children-09-01597-t002:** Birth weight, D_2_O and BMI z-score percentage score for the total group and by sex.

	Total Group (*n* = 91)	Boys (*n* = 40)	Girls (*n* = 51)	
**Birth weight categories**	**N (%)**	**N (%)**	**N (%)**	***χ*^2^ test for sex difference**
LBW	12 (13)	5 (12)	7 (14)	
NBW	78 (87)	35 (88)	44 (86)	0.864
**D_2_O fat percentage categories**				
Underweight	4 (4)	2 (5)	2 (3.9)	
Normal weight	53 (58)	30 (75)	23 (45.1)	0.016
Overweight	19 (21)	6 (15)	13 (25.5)	
Obese	15 (17)	2 (5)	13 (25.5)	
**BMI z-score categories**				
Grades 1 and 2 thinness	7 (8)	3 (7)	4 (8)	
Normal	67 (74)	33 (83)	34 (67)	0.58
Overweight	12 (13)	4 (10)	8 (15)	
Obese	5 (5)	0 (0)	5 (10)	

BMI = body mass index; D_2_O = deuterium oxide; HBW = high birth weight; LBW = low birth weight; NBW = normal birth weight; *χ*^2^ test = Pearson Chi-square *t*-test.

**Table 3 children-09-01597-t003:** Participants’ body composition according to BW groups.

	LBW Categories (*n* = 12)	NBW Group (*n* = 79)	*p*-Value
	Mean ± SD	Mean ± SD	
Age (years)	7.45 ± 0.9	7.71 ± 0.7	0.38
Birth weight (kg)	2.10 ± 0.40	3.20 ± 0.38	<0.0001 *
Weight (kg)	22.27 ± 6.5	25.48 ± 5.8	0.1
Height (cm)	117.42 ± 10.6	122.97 ± 6.1	0.01
FFM (kg; D_2_O)	16.47 ± 4.3	17.95 ± 3.0	0.27
FM (kg; D_2_O)	5.81 ± 2.8	7.53 ± 3.6	0.08
FM (%; D_2_O)	25.59 ± 6.22	28.51 ± 7.27	0.16

* *p*-value of birth weight group differences; independent samples *t*-test; BW = birth weight; cm = centimetres; FFM = fat free mass; FM = fat mass; FM(%) = fat mass percentage; kg = kilograms; D_2_O = deuterium oxide; LBW = low birth weight; NBW = normal birth weight; SD = standard deviation. “FM (kg)”, the statistical test was performed for the data with the logarithmic transformation.

**Table 4 children-09-01597-t004:** Association between BW and body composition measures unadjusted and then adjusted for age, sex, and selected SES variables *.

		BW Unadjusted	BW Adjusted for Age and Sex	BW Adjusted for Age, Sex, and SES
Weight	r	0.23	0.20	0.21
	*p*	0.03 *	0.06	0.20
Height	r	0.33	0.31	0.22
	*p*	<0.001 **	0.003	0.16
FFM (kg; D_2_O)	r	0.27	0.21	0.27
	*p*	0.01 *	0.51	0.10
FM (%; D_2_O)	r	0.11	0.16	0.09
	*p*	0.30	0.12	0.57
FM (kg; D_2_O)	r	0.23	0.16	0.19
	*p*	0.03 *	0.14	0.24

BW = birth weight; FFM = fat free mass; FM = fat mass; FM(%) = fat mass percentage; D_2_O = deuterium oxide; kg = kilograms; *p*-value = independent samples *t*-test; SES = education of the parents, household income, * *p* = 0.05; ** *p* = 0.01.

**Table 5 children-09-01597-t005:** Multiple regression analyses of correlates of BW (independent variable) and FFM, FM, and FM% (dependent variables).

	Dependent Variable	Unstandardised β	Adjusted r Square	*p* Value	95% CI
**Crude models**					
Total group	FFM (kg; D_2_O)	0.24	0.057	0.01	0.032; 0.441
	FM (kg; D_2_O)	0.21	0.044	0.02	0.001; 0.412
	FM (%; D_2_O)	0.18	0.033	0.30	−0.027; 0.338
Boys	FFM (kg; D_2_O)	0.13	0.430	0.43	−0.198; 0.454
	FM (kg; D_2_O)	0.24	0.057	0.08	−0.081; 0.557
	FM (%;D_2_O)	0.25	0.063	0.12	−0.067; 0.568
Girls	FFM (kg; D_2_O)	0.32	0.085	0.02	0.050; 0.574
	FM (kg; D_2_O)	0.18	0.202	0.06	−0.101; 0.464
	FM (%;D_2_O)	0.13	0016	0.38	−0.159; 0.411
**Adjusted models ***					
Total group	FFM (kg; D_2_O)	0.18	0.390	0.07	−0.013; 0.381
	FM (kg; D_2_O)	0.17	0.110	0.04	0.031; 0.375
	FM (%; D_2_O)	0.17	0.075	0.17	-0.069; 0.405
Boys	FFM (kg; D_2_O)	0.24	0.320	0.16	−0.102; 0.576
	FM (kg; D_2_O)	0.27	0.160	0.15	−0.108; 0.653
	FM (%; D_2_O)	0.22	0.094	0.26	−0.171; 0.607
Girls	FFM (kg; D_2_O)	0.12	0.510	0.36	−0.124; 0.381
	FM (kg; D_2_O)	0.12	0.100	0.37	−0.158;0.408
	FM (%; D_2_O)	0.15	0.050	0.37	−0.182; 0.477

* Adjusted for age, sex, and household income; FFM = Fat free mass; FM = fat mass; FM % = fat mass percentage; kg = kilograms, D_2_O = deuterium oxide; 95% CI = 95% confidence interval.

## Data Availability

The datasets used for analyses during the current study are not publicly available due ethical restrictions and participant confidentiality, but are available from the corresponding author on reasonable request and in accordance with the NWU data sharing policy.

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
