# Peer review of "Birth Weight and Body Composition as Determined by Isotopic Dilution with Deuterium Oxide in 6- to 8-Year-Old South African Childrenâ€"

_children, 2022, doi:10.3390/children9101597_

Round 1
Reviewer 1 Report
Dear All, I want to congratulate you for the present study. I believe that after a revision it would be an interesting read, on the topic of childhood obesity assesment and management.
Please review the english language as there are some paragraphs that are not clear to read ad understand. Some examples would be: paragraphs 41-42 that I believe should be reformulated. In paragraphs 57-58, I believe that the authors meant: is of paramount importance. I would also suggest reformulating the hypothesis so that it is clear and concise.
Throughout the study, was BW adjusted for gestational age? It is very important when defining low- normal- and high- BW.
The study design and methods should be more detailed regarding body component measuremet/analysis. Although the method is already published elswhere, I believe that a short recap would make the current study easier to follow and understand.
There are some inconsistencies in the Discussions. Please explain a little more in detail how come the BW is positively correlated to both FFM and FM, but no significantly lower FFM in the LBW group, or significant differences in FFM and FM between the LBW and NBW groups, were found.
Author Response
Reviewer #1
Comments and Suggestions for Authors
Dear All, I want to congratulate you for the present study. I believe that after a revision it would be an interesting read, on the topic of childhood obesity assesment and management.
Response: Thank you for your positive comments.
Comments:
Please review the english language as there are some paragraphs that are not clear to read ad understand. Some examples would be: paragraphs 41-42 that I believe should be reformulated. In paragraphs 57-58, I believe that the authors meant: is of paramount importance. I would also suggest reformulating the hypothesis so that it is clear and concise.
Response: In terms of English language, language editing has been performed in our revised manuscript. In our revised manuscript these sections are revised. Our hypothesis has been amended in our revised manuscript.
Comments:
Throughout the study, was BW adjusted for gestational age? It is very important when defining low- normal- and high- BW.
Response: In terms of adjustment for gestational age, this was not possible because no information about gestational age was recorded in the children’s Road-to-Health records. As such, the lack of data on gestational age is the limitation of the study and we included such information in the limitations of the study. In table 4, we performed analysis adjusted for the current age of the child, as well as for sex and socio-economic status (SES).
For example, in a study Boubred F et al (2020) reported that neighbourhood SES could be considered an important factor for clinicians to better identify mothers at risk of having LGA births in addition to well-known risk factors such as maternal diabetes, obesity and age. Additionally, Bilsteen JF et al (2018), on a Danish Population-Based Study revealed that shorter gestational duration even within the term range was associated with poorer socioeconomic outcomes in adulthood. While adults born at 35 to 38 weeks of gestation experienced only slightly increased risk of adverse socioeconomic outcomes, this may have a significant impact on public health, since a large proportion of all children are born in these weeks. In a systematic review by Belbasis L et al (2016), in which they noted the important association between gestational age and BW, the authors found no statistically significant difference in the summary effect between the studies adjusting for gestational age and the unadjusted studies.
In relation to sex, Catalano PM et al (1995) reported that male neonates had higher FFM than females, which corresponds with the well-recognized sex differences in body composition during adolescence and adult life.
Comments:
The study design and methods should be more detailed regarding body component measuremet/analysis. Although the method is already published elswhere, I believe that a short recap would make the current study easier to follow and understand.
Response: Your comments are noted, and a short recap has been included where necessary.
Comments:
There are some inconsistencies in the Discussions. Please explain a little more in detail how come the BW is positively correlated to both FFM and FM, but no significantly lower FFM in the LBW group, or significant differences in FFM and FM between the LBW and NBW groups, were found.
Response: In our revised manuscript in addition to what we already discussed we provided explanations as ‘FFM and FM were not significantly smaller in the LBW group compared to the NBW group (Table 3), but there was a trend of smaller FM in the LBW group (p=0.08). The small number of children in the LBW group (n=12) is probably the reason why statistically significant differences could not be detected. Also, it is noteworthy to mention that sex was the strongest predictor of FM in this age group.’

Reviewer 2 Report
Congratulations to the authors for a great job. I did not find any errors, except for small editing errors, both the selection of the group, its characteristics, and the description of the methodology used are in my opinion without any reservations. My only comment concerns a discussion that could be discussed more broadly.
Author Response
Reviewer #2
Comments and Suggestions for Authors
Congratulations to the authors for a great job. I did not find any errors, except for small editing errors, both the selection of the group, its characteristics, and the description of the methodology used are in my opinion without any reservations. My only comment concerns a discussion that could be discussed more broadly.
Response:
Thank you for your positive comments, we addressed your raised minor comments in our revised manuscript.

Round 2
Reviewer 1 Report
A very interesting research, that is of scientific value. I now recommend publishing it in the current form.